# Ropivacaine Administration Suppressed A549 Lung Adenocarcinoma Cell Proliferation and Migration via ACE2 Upregulation and Inhibition of the Wnt1 Pathway

**DOI:** 10.3390/ijms25179334

**Published:** 2024-08-28

**Authors:** Masae Iwasaki, Makiko Yamamoto, Masahiro Tomihari, Masashi Ishikawa

**Affiliations:** Department of Anesthesiology and Pain Medicine, Graduate School of Medicine, Nippon Medical School, Tokyo 113-8602, Japan; m-yamamoto@nms.ac.jp (M.Y.); m-tomihari@nms.ac.jp (M.T.); masashi-i@nms.ac.jp (M.I.)

**Keywords:** ropivacaine, ACE2, HIF1α, Wnt1 pathway, PCR array

## Abstract

Background: Previous studies have suggested that perioperative anesthesia could have direct impacts on cancer cell biology. The present study investigated the effects of ropivacaine administration on lung adenocarcinoma cells. Methods: Ropivacaine was administered to A549 cells at concentrations of 0.1, 1, and 6 mM for 2 h. Angiotensin-converting enzyme 2 (ACE2) small interfering RNA (siRNA) transfection was performed 6 h prior to ropivacaine administration. Cell proliferation and migration were assessed with cell counting kit 8 (CCK-8) and a wound healing assay at 0 and 24 h after anesthesia exposure. PCR arrays were performed, followed by PCR validation. Results: Ropivacaine administration inhibited A549 cell proliferation and migration in a concentration-dependent manner, with ACE2 upregulation and HIF1α (hypoxia-inducible factor 1α) downregulation. The anticancer effect of ropivacaine was canceled out via ACE2 siRNA transfection. PCR arrays showed specific gene change patterns in the ropivacaine and respective ACE2-knockdown groups. EGFR (epidermal growth factor receptor), BAX (Bcl-2-associated X protein) and BCL2 (B-cell/CLL lymphoma 2) were suppressed with ropivacaine administration; these effects were reversed via ACE2 siRNA induction. Conclusion: Ropivacaine administration inhibited A549 cell biology in conjunction with ACE2 upregulation via the inhibition of the Wnt1 (wingless/Integrated 1) pathway.

## 1. Introduction

The prevalence of lung cancer continues to increase worldwide, with the estimated number of patients reaching 2.2 million in 2024 [1]. According to the Japanese clinical guideline for lung cancers, the first-line treatment for lung cancers is surgical removal under general anesthesia, combined with chemotherapy or radiation therapy [2]. Statistically, the prognosis for lung cancer patients is worse than that for other cancers; the five-year survival rate for non-small-cell lung cancer patients is less than 40% in Japan [2].

The postoperative recurrence of cancer significantly affects the one-year survival rate for all types of cancers. It has been speculated that intraoperative anesthetics could cause changes in cancer cell biology via gene expression changes in residual cancer cells after oncosurgery, leading to postoperative recurrence [3]. On the other hand, in vitro studies have shown that propofol can suppress cancer progression and proliferation in lung cancer cells [4,5,6]. This paradox has led to numerous studies being conducted over the last decade comparing the clinical effects of intravenous anesthesia using propofol with those of inhalation anesthesia using other agents. In a retrospective study of more than 7000 patients who underwent oncosurgery, mortality at 3 years after surgery was approximately 50% greater with inhalation anesthesia than with intravenous anesthesia [7]. In patients undergoing mastectomy, intraoperative anesthesia with ketorolac was found to decrease the risk of breast cancer relapse compared with the intraoperative use of other analgesics [8]. Among patients with lung cancer, the most common cancer globally, several clinical datasets have shown a slight advantage of intravenous anesthesia over inhalation anesthesia in terms of cancer recurrence and mortality. One meta-analysis and one clinical retrospective study showed that intravenous anesthesia might be more beneficial at reducing cancer recurrence and mortality compared to inhalation anesthesia [7,9]. Other retrospective studies have shown no differences in mortality between intravenous and inhalation anesthesia methods [10,11]. Also, a narrative review suggested that there was no advantage of total intravenous anesthesia over inhalation anesthesia in reducing mortality [12]. The direct effects of anesthetics on cancer cell biology have been revealed in several research papers in vitro and in vivo, but the gap between the clinical data and the subclinical data is still wide.

There have been a few studies on the associations of local anesthetics with cancer biology changes [3,13,14,15,16]. In clinical practice, regional anesthesia may suppress cancer progression via the control of surgical stress [3]. One prospective clinical study showed that using regional anesthetics for lung cancer surgery could be equally beneficial in terms of cancer-free survival compared to using general anesthesia only [13]. In an in vivo study, lidocaine intravenous injection combined with cisplatin therapy suppressed cancer growth in a xenograft mouse model [14]. Also, lidocaine administration was reported to suppress lung cancer invasion in vitro [17]. Ropivacaine, one of the common local anesthetics, might suppress cancer progression in vitro at doses of 0.5, 1 and 2 mM for 24 and 48 h [16]. The micromechanisms of the direct effects of local anesthetics remain unclear.

The expressions of angiotensin-converting enzyme 2 (ACE2) and hypoxia-inducible factor 1 α (HIF1α) in cancer tissue are known as predictive markers of cancer recurrence. HIF1α expression in clinicopathological samples is well known as a poor prognostic factor in many types of human cancers, including lung cancer [18]. Suppression of HIF1α has been shown to improve the efficacy of radiation therapy for lung adenocarcinoma [19]. Interestingly, a previous in vitro study showed that HIF1α expression in various types of cancer cells changed after anesthesia [20,21,22].

ACE2 has multiple roles via Mas receptors in the human body, including roles in fighting inflammation and protecting various organs [23]. ACE2 is considered a good prognostic factor in several types of cancers; its upregulation is associated with favorable survival [24]. A prospective observational study of lung cancer revealed that ACE2 expression in pathological samples was a predictor of better outcomes [25]. ACE2 directly inhibited cancer angiogenesis, cell growth and VEGFA (vascular endothelial growth factor A) expressions in A549 in vitro and in vivo [26].

The interaction between HIF1α and ACE2 as well as any changes in ACE2 expression after anesthetic administration remain unclear. The present study aimed to clarify the impact of anesthetics on tumor prognosis. Our hypothesis was that changes in ACE2 expression following ropivacaine administration might induce changes in cancer cell biology via HIF1α and other pathways. Since the most common histological type of clinical lung cancer is adenocarcinoma [27], we selected adenocarcinoma cell type A549 for our in vitro investigation.

## 2. Results

### 2.1. Ropivacaine Suppressed A549 Cell Proliferation and Migration in a Dose-Dependent Manner; This Effect Was Reversed with ACE2 siRNA Transfection

#### 2.1.1. Cell Proliferation Test

Cells were treated with each concentration of ropivacaine with/without siRNA transfection. The treatments were defined as follows: C, no medication; R0.1, ropivacaine at 0.1 mM for 2 h; R1, ropivacaine at 1 mM for 2 h; R6, ropivacaine at 6 mM for 2; and si, siRNA transfection for 6 h prior to ropivacaine administration.

Cell proliferation was suppressed via ropivacaine administration in a dose-dependent manner, as shown as Figure 1 (mean ± SD, relative ratio to C, *n* = 6; C 1.000 ± 0.060 vs. R0.1 0.831 ± 0.134 vs. R1 0.752 ± 0.072 vs. R6, 0.603 ± 0.070; C vs. R 0.1, *p* = 0.008; C vs. R1, *p* = 0.000; C vs. R6, *p* = 0.000; R0.1 vs. R1, *p* = 0.740; R0.1 vs. R6, *p* = 0.000; R1 vs. R6, *p* = 0.030).

The knockdown efficiency of 6 h siRNA transfection was calculated via qRT-PCR; the efficiency was 65.4% for siRNA1 and 55.3% for siRNA2. The cells transfected with ACE siRNA1 showed no significant difference in cell proliferation compared to the cells in the siRNA-untreated groups, except for a significant increase in the siRNA1+R6 group (R6 0.603 ± 0.070 vs. siRNA1+R6 0.802 ± 0.062; *p* = 0.001). With ACE2 siRNA2 transfection, both the siRNA2+R1 and siRNA2+R6 groups showed significant increases in cell proliferation compared to the respective siRNA-untreated groups (R1 0.752 ± 0.072 vs. siRNA2+R1 0.947 ± 0.067, *p* = 0.002; R6 0.603 ± 0.070 vs. siRNA2+R6 0.859 ± 0.053, *p* = 0.000). Between siRNA 1 and 2, siRNA2 transfection showed a greater increase in cell proliferation with ropivacaine 1 mM (siRNA1+R1 0.780 ± 0.024 vs. siRNA2+R1 0.937 ± 0.067, *p* = 0.017). siRNA2 was therefore used for the subsequent investigation.

#### 2.1.2. Cell Migration

The results of the wound healing assay are shown in Figure 2. Compared to the C group, the R groups showed significantly reduced cell migration in a dose-dependent manner (C 24.97 ± 3.501 vs. R0.1 10.79 ± 3.070 vs. R1 8.907 ± 2.913 vs. R6 2.604 ± 2.072; NC vs. R0.1, *p* = 0.000; NC vs. R1, *p* = 0.000; NC vs. R, *p* = 0.000; R0.1 vs. R1, *p* = 0.976; R0.1 vs. R6, *p* = 0.003; R1 vs. R6, *p* = 0.430). After siRNA2 transfection, there was a further decrease in cell migration in the siRNA2+ R0.1 group compared to the siRNA-untreated group (R0.1 10.79 ± 3.070 vs. siRNA2+R0.1 3.983 ± 2.088, *p* = 0.022).

### 2.2. Cluster Analysis of PCR Array Showed Specific Gene Expression Changes with Ropivacaine Administration and ACE2 siRNA Transfection

#### 2.2.1. PCR Array and Cluster Analysis

The cluster analysis using Taqman Array Human Molecular Mechanisms of Cancer showed clear differences between ropivacaine administration with and without ACE2 siRNA transfection (Figure 3). Also, the cluster analysis indicated that the WNT1 gene showed the most radical expression changes with ropivacaine administration.

Among the 84 genes examined via the PCR array, 18 genes were changed via ropivacaine administration (13 upregulated, 5 downregulated), and these are listed in Supplemental Appendix A. Eleven genes were changed via ACE2 siRNA transfection without ropivacaine administration (six upregulated, five downregulated). With ropivacaine administration, 10 genes varied depending on ACE2 siRNA transfection (4 upregulated, 6 downregulated by transfection). The WNT1 gene showed the greatest alteration in ropivacaine-induced expression with/without ACE2 siRNA transfection. Therefore, the components of the Wnt1 pathway, namely the BAX, BCL2 and WNT1 genes, were selected for qRT-PCR validation to confirm the results of the PCR array.

#### 2.2.2. qRT-PCR Confirmed the Efficacy of ACE2 siRNA Transfection and Its Effects on Wnt1 Pathway Genes

The results of qRT-PCR are shown in Figure 4a–f; numerical data are listed in Appendix A. ACE2 siRNA2 transfection inhibited ACE2 expression to 40% of the level in the siRNA-untreated groups, confirming the efficacy of the designed ACE2 siRNA and the induction procedure (Figure 4a; C vs. siRNA, *p* = 0.000; R0.1 vs. siRNA+R0.1, *p* = 0.000; R1 vs. siRNA+R1, *p* = 0.000). Among the siRNA-treated groups, there were no significant changes in ACE2 expression between the control and those treated with any ropivacaine concentration (siRNA only vs. siRNA+R0.1, *p* = 0.999; siRNA only vs. siRNA+R1, *p* = 0.999; siRNA vs. siRNA+R6, *p* = 0.999; siRNA+R0.1 vs. siRNA+R1, *p* = 0.999; siRNA+R0.1 vs. siRNA+R6, *p* = 0.999; siRNA+R1 vs. siRNA+R6, *p* = 0.822).

Figure 4b shows the changes in BAX gene expression after ropivacaine administration with/without ACE2 siRNA transfection, with significant differences between the R1 group and siRNA+R1 group and between the R6 group and siRNA+R6 group (R1 vs. siRNA+R1, *p* = 0.007; R6 vs. R6+siRNA, *p* = 0.021). Regarding BCL2 gene expression, there were significant differences only between the R1 group and the R6 group among the siRNA-untreated groups (Figure 4c, R1 vs. R6, *p* = 0.021). Meanwhile, the siRNA+R0.1 group (but not the siRNA-treated groups with R1 or R6) showed decreased BCL2 expressions compared to the siRNA-only group (siRNA vs. siRNA+R0.1, *p* = 0.000; siRNA vs. siRNA+R1, *p* = 0.105; siRNA vs. siRNA+R6, *p* = 0.081). The siRNA+R0.1 group and the siRNA+R1 group showed decreased BCL2 expressions compared to the respective siRNA-untreated groups (R0.1 vs. siRNA+R0.1, *p* = 0.000; R1 vs. siRNA+R1, *p* = 0.001). EGFR gene expression increased in the siRNA+R1 group and the siRNA+R6 group compared to that in the respective siRNA-untreated groups (Figure 4d, R1 vs. siRNA+R1, *p* = 0.027; R6 vs. siRNA+R6, *p* = 0.002). HIF1α expression was lower in the R6 group than that in the C group, and lower in the siRNA+R1 group than that in the R1 group (Figure 4e, C vs. R6, *p* = 0.020; R1 vs. siRNA+R1, *p* = 0.023). The WNT1 gene showed drastic changes with ACE2 siRNA transfection alone, but no changes among R groups or between siRNA-treated and -untreated pairs with R administration (Figure 4f, C vs. siRNA, *p* = 0.000; R0.1 vs. siRNA+R0.1, *p* = 0.001; R1 vs. siRNA+R1, *p* = 0.000; R0.1 vs. siRNA+R0.1, *p* = 0.000; R6 vs. siRNA+R6, *p* = 0.994).

#### 2.2.3. The PCR Array and qRT-PCR Were Well Correlated

Figure 5 compares the results of the PCR array and qRT-PCR using Bland–Altman analysis. As shown in panels e and g, the differences are plotted within the limits of agreements, indicating that these comparisons contained only random errors. Panels f and h show that systematic errors may have existed between the PCR array and qRT-PCR. All the numerical data of the PCR array and qRT-PCR are listed in Appendix A.

### 2.3. Immunofluorescent Study Confirmed That Ropivacaine Administration Enhanced ACE2 Expression Dose-Dependently with/without ACE2 siRNA Transfection

#### 2.3.1. ACE2 Expression

The representative immunofluorescence images and analysis of A549 cells after ropivacaine administration with and without ACE2 siRNA transfection are shown in Figure 6 and Figure 7. Figure 6 shows that ACE2 expression decreased with ACE2 siRNA transfection compared to that in the siRNA-untreated groups, confirming the efficacy of ACE2 siRNA transfection. Ropivacaine administration enhanced ACE2 expression dose-dependently with/without ACE2 siRNA transfection.

#### 2.3.2. Cancer Malignancy Markers

Figure 7 shows representative immunofluorescence images of cancer malignancy and/or Wnt1 pathway biomarkers. The statistical analysis of immunofluorescence intensity showed that ropivacaine administration significantly decreased the expressions of HIF1α, MMP9 and β catenin, but these results were drastically reversed with ACE2 siRNA transfection. β catenin, one of the cancer malignancy markers in the Wnt1 pathway, showed the most drastic changes in this regard.

## 3. Discussion

The present study investigated the effects of ropivacaine and its potential anti-cancer mechanism using human lung adenocarcinoma A549 and ACE2 siRNA transfection. Ropivacaine administration suppressed A549 cell migration and proliferation in a dose-dependent manner, and this suppression was reversed with ACE2 siRNA transfection. ACE2 siRNA transfection decreased the HIF1α, MMP9 and Wnt1 pathway biomarkers regardless of ropivacaine administration, indicating that ropivacaine restrains A549 cell biology via ACE2 upregulation.

Ropivacaine is one of the most widely used local anesthetics, but there is little research about its direct effects on cancer cell biology. In vitro ropivacaine administration was reported to suppress A549 cell malignancy in two previous studies [16,28]. These reports used a similar concentration of ropivacaine to that used in the present study, with a longer exposure time of 24–48 h, which did not seem realistic for a clinical setting. The present study demonstrated that 2 h was a sufficient exposure duration for revealing the anti-cancer effects of ropivacaine (Figure 1 and Figure 2). HIF1α, MMP9 and β catenin are well known as cancer malignancy markers. The present data indicate that ropivacaine administration could suppress cancer progression directly (Figure 7).

Also, our data revealed that ropivacaine suppressed HIF1α, MMP9 and β catenin via ACE2 upregulation. A high intracellular ACE2 level is considered a good prognostic factor, and has been reported to inhibit cancer angiogenesis and cell growth both in vitro and in vivo [26]. As for the macromechanism, ACE2 has been suggested to protect multiple organs against inflammation [29,30]. ACE is known as a functional receptor of SARS viruses including SARS-no-CoV2, and the severity of COVID-19 has been associated with the levels of ACE2 and TMPRSS2 co-expression and the proteinase activity of FURIN [23]. One clinical meta-analysis showed that ACE2 expressions in lung cancer and other cancers were higher than those in the lungs of patients with COVID-19 [31]. However, this research also showed that ACE2 in lung cancer had little interaction with infectious diseases or inflammation pathways, indicating that the severity of a cytokine storm may be directly associated with COVID-19 severity and mortality. Another clinical investigation showed that lung cancer patients might be more susceptible to SARS-CoV-2 infection than patients without cancer [32]. On the other hand, a previous study on the serum ACE2 levels of postoperative lung cancer patients showed that lower serum ACE2 was significantly associated with pneumonia, pleural effusion and higher mortality [25]. Taken together, these results indicate that the high ACE2 expression in lung cancer might lead to a higher risk of SARS-CoV-2 entry but not a higher risk of the severity of COVID-19, and should be a safe therapy target.

The possible linkage between ACE2 and HIF1α in lung cancer is a new discovery, and has only been suggested in two previous studies [33,34]. One study showed that higher ACE2 expression in clinical pathological tissue was associated with worse chemoresistance in breast cancer, with a connection to the ROS–AKT–HIF1α axis [33]. The other showed that ACE2 and HIF1α balanced each other in infantile hemangioma in vitro [34]. The present data on ACE2 siRNA transfection demonstrated the downregulation by ACE2 of HIF1α, MMP9 and β catenin in A549 cells (Figure 6 and Figure 7), suggesting that the interaction between ACE2 and HIF1α and other cancer-promoting genes could differ depending on the cancer type.

The siRNA treatment was performed for 6 h, and cells were exposed to ropivacaine for 2 h, with an RNA study carried out at 6 h after ropivacaine exposure and immunofluorescent analysis carried out at 24 h after ropivacaine exposure. The results of our PCR array and qRT-PCR showed the knockdown effect of ACE2 siRNA transfection (Figure 3 and Figure 4). The immunofluorescence study showed that ACE2 proteins were increased even in the siRNA-treated groups, indicating that a resting time longer than 24 h might be needed to achieve a full knockdown effect. Even with a 24 h resting time, we still observed a significant decrease in siRNA-treated groups compared to the C group.

## 4. Materials and Methods

### 4.1. Cell Culture

The human-authenticated lung adenocarcinoma cell line, A549 (RIKEN BioResource Research Center, Kyoto, Japan), was cultured in RPMI 1640 (Thermo Fisher Scientific, Tokyo, Japan) containing 10% fetal bovine serum and 1% penicillin/streptomycin (Thermo Fisher Scientific) and maintained in a humidified incubator at 37 °C with a 5% CO_2_ atmosphere.

### 4.2. Anesthetic Administration

Ropivacaine (the R group) was obtained from Astra Zeneca (Tokyo, Japan). The concentrations administered to cell cultures were selected via reference to a previous study [16], namely 0.1, 1, and 6 mM. Our preliminary experiments showed that the solution containing over 6 mM of ropivacaine underwent crystallization, and thus, we chose a maximum dose of 6 mM for use in the present study. The controls (C group) received no ropivacaine. Cells were exposed to the medication for 2 h, a reasonable approximation of lung cancer surgery duration.

### 4.3. siRNA Transfection

siRNA transfection was performed according to a previous study [35]. Briefly, cells were transfected with scrambled siRNA (ScrRNA; the C group) or one of two different ACE2 siRNA constructs at a concentration of 20 nmol/L (Appendix A; Merck Chemicals B.V., Tokyo, Japan; Ajinomoto Bio-Pharma., Osaka, Japan). The transfection was facilitated with HiPerfect Transfection Reagent (Qiagen, Tokyo, Japan). After 6 h of transfection, the solution was replaced with standard culture medium. Then, the siRNA-treated cells were subjected to ropivacaine administration for the immunostaining and RNA study.

### 4.4. Cell Proliferation Test (Cell Counting Kit-8, CCK-8 Assay)

Cells were plated at approximate populations of 5 × 10^3^ per well on 96-well plates and allowed to adhere for 24 h. The cells were then exposed to each concentration of ropivacaine for 2 h and allowed to sit for another 24 h. The CCK-8 proliferation test was performed in accordance with the manufacturer’s manual (Dojindo Laboratories., Kumamoto, Japan) using a SpectraMax i3x microplate reader (Molecular Devices, LLC., Tokyo, Japan).

### 4.5. Wound-Healing Assay

Cells were plated at 3 × 10^5^ per well of a 3-well insert (Culture-Insert 3 well; ibidi GmbH; Fitchburg, WI, USA) on a 35 mm Petri dish, and allowed to rest for 24 h. The cells were exposed to each concentration of ropivacaine for 2 h; then, the insert was removed and the cells were left in the standard medium for another 24 h. The gap closure was examined with an optical microscope (CKX31; Olympus, Tokyo, Japan) at a magnification of 20× (LCAch N; Olympus). Images were assessed with Image J 1.54i (National Institutes of Health, Bethesda, MD, USA) to calculate the gap closure rate.

### 4.6. RNA Extraction

Total RNA was extracted from confluent cells on 60 mm Petri dishes at 6 h after 2 h of ropivacaine exposure using RNeasy Mini Kit^®^ and QIAshredder (Qiagen) according to the manufacturer’s instructions. RNA quantity and quality were assessed using NanoDrop (Thermo Fisher Scientific). Samples with an A260/A280 ratio > 1.8 were considered to be of sufficient quality for further analysis. An amount of 1 ug of total RNA samples was converted into cDNA using a high-capacity cDNA reverse transcription kit (Thermo Fisher Scientific).

### 4.7. PCR Array

PCR array analysis was performed and analyzed using TaqMan Array Human Molecular Mechanisms of Cancer and QuantStudio^®^ 5 (Thermo Fisher Scientific) following the manufacturer’s protocol. Glyceraldehyde-3-phosphate dehydrogenase (GAPDH) mRNA was selected as the endogenous control to evaluate each relative expression ratio, using the comparative 2^−ΔΔCT^ method. PCR array analysis was performed using ExpressionSuite v1.3 Software (Thermo Fisher Scientific), and cluster analysis determined the average linkage between clusters based on Euclidian distance.

### 4.8. Validation PCR and qRT-PCR

Some representative genes (BAX, BCL2, EGFR and WNT1) were subjected to further qRT-PCR or validation PCR using TaqMan primers and TaqMan Fast Advanced Mastermix (Thermo Fisher Scientific). The designed primer sequences for ACE2 and HIF1α with r^2^ values and efficiencies are listed in Appendix A (Ajinomoto Bio-Pharma). These cDNA samples were mixed with Fast SYBR Green mastermix before qRT-PCR with QuantStudio^®^ 5 (Thermo Fisher Scientific). qRT-PCR analysis was performed using ExpressionSuite v1.3 Software (Thermo Fisher Scientific).

### 4.9. Immunofluorescence Study

At 24 h after seeding the 3 × 10^5^ cells on one 13 mm cover glass per well of a 24-well plate, cells were exposed to ropivacaine at different concentrations for 2 h. Then, the cells were allowed to rest for 24 h. The cells were fixed with 4% paraformaldehyde and blocked with 10% normal donkey serum (Merck Chemicals B.V.) and then incubated overnight at 4 °C with each of the following primary antibodies: rabbit anti-ACE2 (1:200, Abcam plc, Tokyo, Japan), rabbit anti-HIF1α (1:200, Novus Biologicals, LLC, Centennial, CO, USA), rabbit anti-MMP9 (1:200, Cell Signaling Technology, Tokyo, Japan) and rabbit anti-β catenin (1:200, Cell Signaling Technology). The samples were incubated with a conjugated secondary antibody (Alexa Fluor^®^ 568; Thermo Fisher Scientific) and co-stained with Vectashield mounting medium containing DAPI (Thermo Fisher Scientific). Six areas of each slide were randomly selected for imaging under a microscope (BX53; Olympus) at a magnification of 40× (DP74; Olympus), followed by analysis using Image J 1.54i (National Institutes of Health).

### 4.10. Statistical Analysis

All numerical data are presented as scatter dot plots or means ± SDs. We determined that a group size of n = 6 was needed to show a 30% change with 80% power at 5% significance. One-way ANOVA analysis followed by a post hoc Tukey–Kramer test was applied for statistical analysis using Prism 8.0 (GraphPad Software, San Diego, CA, USA). For analysis of the PCR array results, the false discovery rate was set as 0.1 using the program QVALUE 2.0 (http://github.com/jdstorey/qvalue (accessed on 31 March 2024)). To compare the results of the PCR array and RT-PCR, Bland–Altman analysis was performed with Prism 8.0, using the difference from the average. In all experiments, *p* values < 0.05 were considered to indicate significance.

## 5. Conclusions

Ropivacaine administration inhibited A549 cell biology in conjunction with ACE2 upregulation via the inhibition of the Wnt1 (wingless/Integrated 1) pathway.

The present study contains some limitations. First, this was an in vitro study culturing cells of a single type with a fixed anesthetic administration protocol. A study with several cell lines, multiple exposure times, and single/multiple administration would provide further insights into the effects of ropivacaine on cancer cell biology. Animal and/or clinical studies will provide more profitable evidence when the timing is appropriate. Second, there may be more knockdown points to focus on using siRNA transfection.

These limitations notwithstanding, the present study provided direct evidence that upregulation of ACE2 expression in A549 cells is a potential therapeutic target, improving the prospects for lung adenocarcinoma. Further in vitro investigations and optimization of clinical settings will be needed.

## Figures and Tables

**Figure 1 ijms-25-09334-f001:**
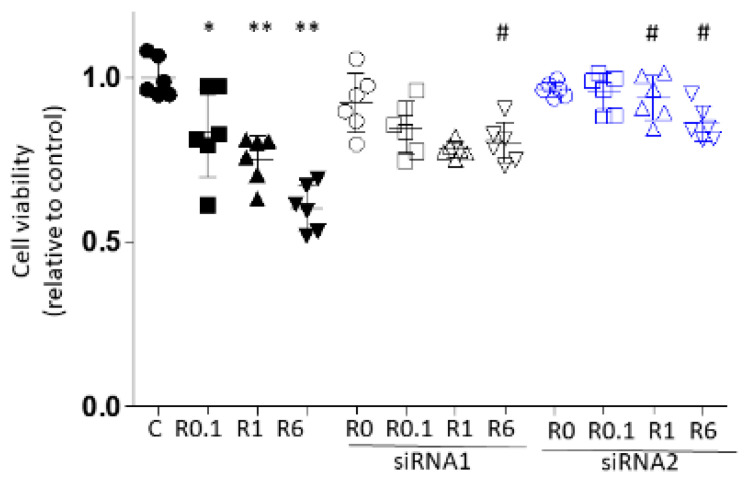
A549 cell proliferation analysis using CCK8 at 24 h after 2 h of ropivacaine administration with/without ACE2 siRNA1/2 transfection. *, *p* < 0.05 compared to the C group; **, *p* < 0.01 compared to the C group; #, *p* < 0.05 compared to the siRNA-untreated group; n = 6, one-way ANOVA followed by a post hoc Tukey test. Dot: no administration of ropivacaine, square: 0.1 µM ropivacaine, triangle: 1 µM ropivacaine, downward triangle: 6 µM ropivacaine, black: no siRNA transfection, black and white: with siRNA1 transfection, blue and white: with siRNA2 transfection, C, control (scrRNA-treated; R, ropivacaine; CCK8, cell count kit 8.

**Figure 2 ijms-25-09334-f002:**
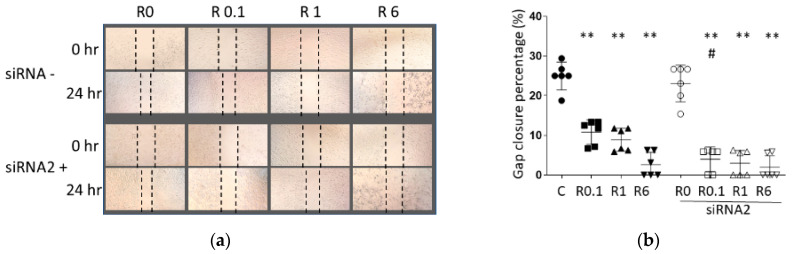
A549 cell migration analysis with wound healing assay after ropivacaine administration with/without ACE2 siRNA2 transfection. (**a**) Representative images of the wound healing assay with each investigation; (**b**) analysis of gap closure percentage at 24 h after ropivacaine administration. **, *p* < 0.01 compared to the C group; #, *p* < 0.05 compared to the siRNA-untreated group; n = 6, one-way ANOVA followed by a post hoc Tukey test. Dot: no administration of ropivacaine, square: 0.1 µM ropivacaine, triangle: 1 µM ropivacaine, downward triangle: 6 µM ropivacaine, black: no siRNA transfection, black and white: with siRNA1 transfection, C, control (scrRNA-treated); R, ropivacaine.

**Figure 3 ijms-25-09334-f003:**
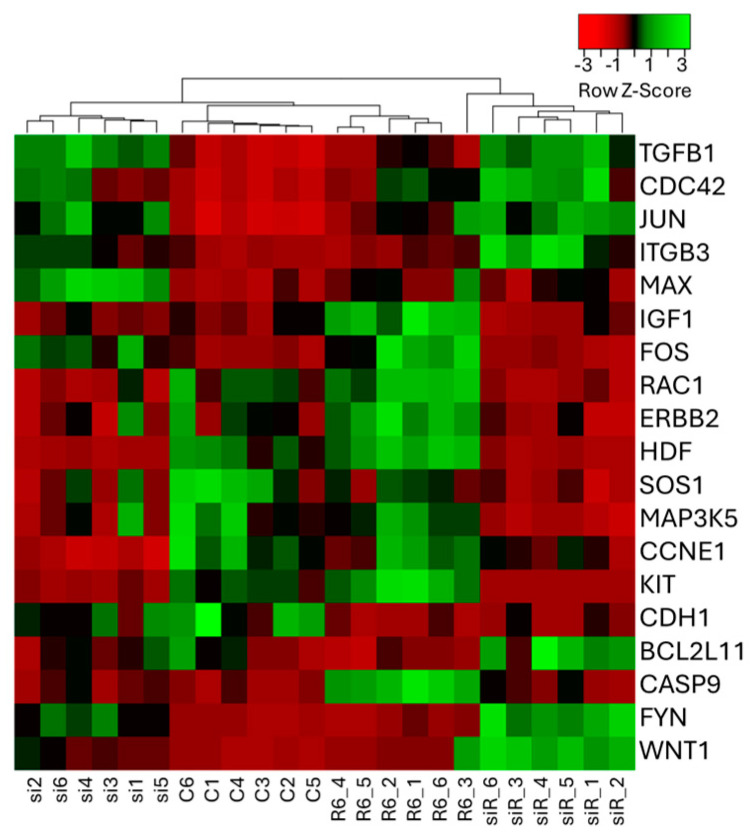
PCR array results of cancer-related genes. A549 cells after ropivacaine administration were analyzed via the PCR array and unsupervised hierarchical cluster analysis with the average linkage and Euclidean dissimilarity methods, and compared to the endogenous control, GAPDH. Red and green colors indicate relatively high and low expressions, respectively (*n* = 6, one-way ANOVA followed by a post hoc Tukey test). C, control (scrRNA-treated); R, ropivacaine; si, siRNA; siR, siRNA+R6; BCL2L11, B-cell/CLL lymphoma 2 ligand 11; CASP9, caspase 9; CCNE1, cyclin E1; CDC42, cell division cycle 42; CDH1, cadherin 1; ERBB2, erb-B2 receptor tyrosine kinase 2; FOS, fos proto-oncogene, AP-1 transcription factor subunit; FYN, FYN proto-oncogene, Src family tyrosine kinase; HDF, hepatocyte growth factor; IGF1, insulin-like growth factor 1; ITGB3, integrin subunit beta 3; JUN, Jun proto-oncogene, AP-1 transcription factor subunit; KIT, KIT proto-oncogene receptor tyrosine kinase; MAP3K5, mitogen-activated protein kinase 5; MAX, MYC-associated factor X; RAC1, Rac family small GTPase 1; SOS1, SOS Ras/Rac guanine nucleotide exchange factor 1; TGFB1, transforming growth factor beta 1; WNT1, wingless/Integrated 1.

**Figure 4 ijms-25-09334-f004:**
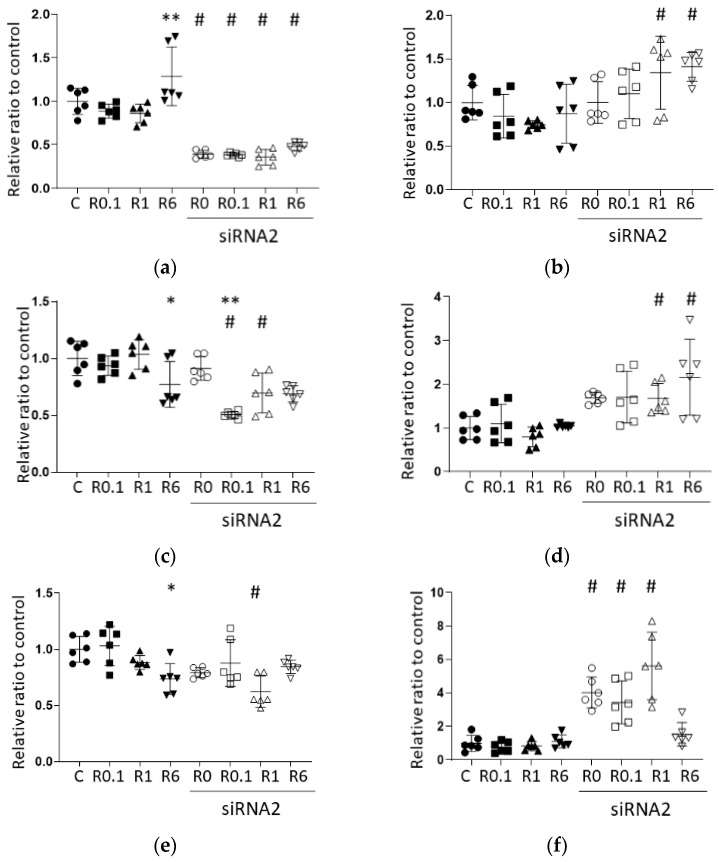
qRT-PCR results of representative genes at 24 h after ropivacaine administration with/without ACE2 siRNA transfection. (**a**) ACE2, (**b**) BAX, (**c**) BCL2, (**d**) EGFR, (**e**) HIF1α and (**f**) WNT1. Data are shown as plots and means ± SDs. *, *p* < 0.05 compared to the C group; **, *p* < 0.01 compared to the C group; #, *p* < 0.05 compared to siRNA-untreated group; *n* = 6, one-way ANOVA followed by a post hoc Tukey test. C. Dot: no administration of ropivacaine, square: 0.1 µM ropivacaine, triangle: 1 µM ropivacaine, downward triangle: 6 µM ropivacaine, black: no siRNA transfection, black and white: with siRNA1 transfection, control (scrRNA-treated); R, ropivacaine; si, siRNA; ACE2, angiotensin-converting enzyme 2; BAX, Bcl-2-associated X protein; BCL2, B-cell/CLL lymphoma 2; EGFR, epidermal growth factor receptor; HIF1α, hypoxia-inducible factor 1 α and WNT1, wingless/Integrated 1.

**Figure 5 ijms-25-09334-f005:**
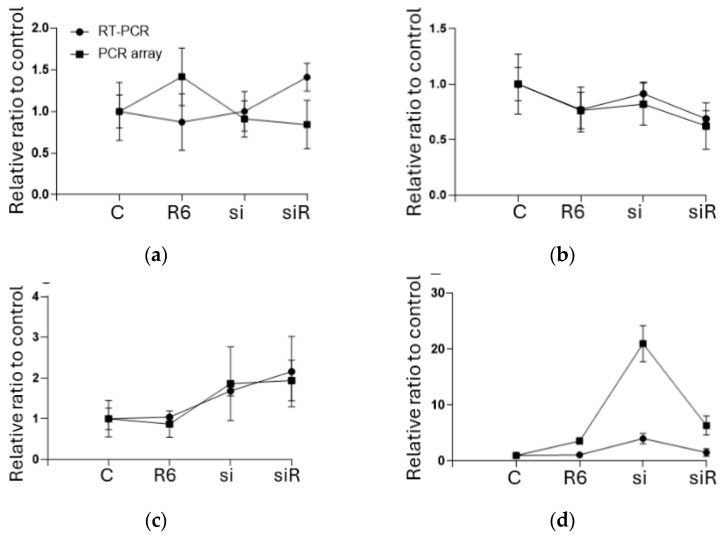
Comparison of the PCR array and qRT-PCR results. (**a**) BAX; (**b**) BCL2; (**c**) EGFR; (**d**) WNT1. (**e**,**f**): Bland-Altman analyses comparing the results of the RT-PCR and PCR array. (**e**) BAX; (**f**) BCL2; (**g**) EGFR; (**h**) WNT1. Data are shown as plots and means ± SDs; *n* = 6, one-way ANOVA followed by a post hoc Tukey test compared to the control group. C, control (scrRNA-treated); R, ropivacaine; si, siRNA; BAX, Bcl-2-associated X protein; BCL2, B-cell/CLL lymphoma 2; EGFR, epidermal growth factor receptor; WNT1, wingless/Integrated 1.

**Figure 6 ijms-25-09334-f006:**
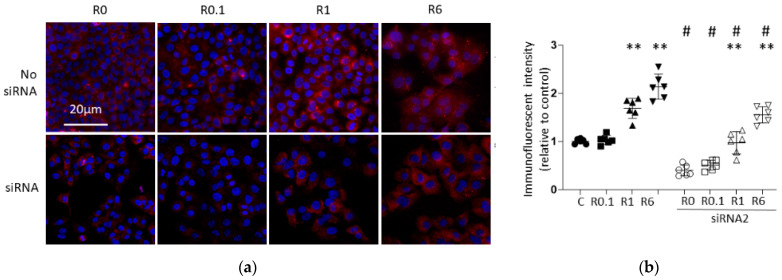
Immunofluorescent images targeting ACE2 (anti-cancer factor). (**a**) Representative immunofluorescent images of A549 cells at 24 h after 2 h of ropivacaine exposure with/without ACE2 siRNA transfection; blue, DAPI; red, ACE2 (scale bar: 20 µm, ×20). (**b**) Analysis of the immunofluorescent intensity. Data are shown as plots and means ± SDs. **, *p* < 0.01 compared to the C group; #, *p* < 0.01 compared to the siRNA-untreated group; n = 6, one-way ANOVA followed by a post hoc Tukey test. Dot: no administration of ropivacaine, square: 0.1 µM ropivacaine, triangle: 1 µM ropivacaine, downward triangle: 6 µM ropivacaine, black: no siRNA transfection, black and white: with siRNA1 transfection, C, control (scrRNA-treated); R, ropivacaine; si, siRNA; ACE2, angiotensin-converting enzyme 2.

**Figure 7 ijms-25-09334-f007:**
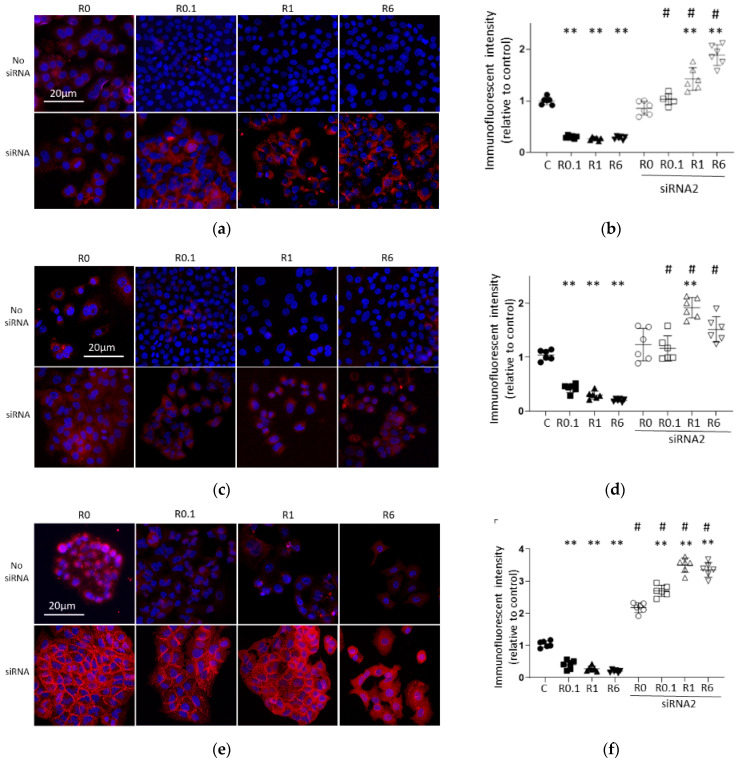
Immunofluorescent images targeting pro-cancer factors. Representative immunofluorescent images of A549 cells at 24 h after 2 h ropivacaine exposure with (**lower**) and without (**upper**) ACE2 siRNA transfection are shown. Blue, DAPI; red, each target marker (scale bar: 20 µm, ×20). (**a**) HIF1α (cancer malignancy marker); (**b**) analysis of the immunofluorescent intensity of HIF1α; (**c**) MMP9 (metastasis marker); (**d**) analysis of the immunofluorescent intensity of MMP9; (**e**) β catenin (cancer malignancy marker); (**f**) analysis of the immunofluorescent intensity of β catenin. Data are shown as plots and means ± SDs. **, *p* < 0.01 compared to the C group; #, *p* < 0.01 compared to siRNA-untreated group; *n* = 6, one-way ANOVA followed by a post hoc Tukey test. Dot: no administration of ropivacaine, square: 0.1 µM ropivacaine, triangle: 1 µM ropivacaine, downward triangle: 6 µM ropivacaine, black: no siRNA transfection, black and white: with siRNA1 transfection, C, control (scrRNA-treated); R, ropivacaine; ACE2, angiotensin-converting enzyme 2; si, siRNA; HIF1α, hypoxia-inducible factor 1α; MMP9, matrix metalloproteinase 9.

## Data Availability

The raw data supporting the conclusions of this article will be made available by the authors on request.

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
