# Peer review of "Ropivacaine Administration Suppressed A549 Lung Adenocarcinoma Cell Proliferation and Migration via ACE2 Upregulation and Inhibition of the Wnt1 Pathway"

_ijms, 2024, doi:10.3390/ijms25179334_

Round 1

Reviewer 1 Report

Comments and Suggestions for Authors In my opinion, the experiments were well designed, but the authors should clarify why anesthetic concentrations of 0.1, 1 and 6 μmol/L were used.  Appropriate methods were used too. As alveolar type 2 (AT2) cells have been identified as major players in a variety of lung diseases, including lung adenocarcinomas, A549 cells are a good model for studying the influence of drugs or anesthetics on lung cancer biology.  All references used are appropriate and used correctly, but the authors should correct errors in their text. Based on the results obtained, a concrete conclusion should be formulated and a further research direction suggested. (The limitations of the study are mentioned in the Discussion chapter and should be moved to the Conclusions chapter) In the introduction section, the authors should cite studies related to this rate: There are few studies on the association between local anesthesia and biological changes in cancer.„ Detailed corrections of the English language are needed. Comments on the Quality of English Language  Detailed corrections of the English language are needed.

Author Response

Dear reviewers,

Thank you for your letter and the reviewers’ comments on our above-titled manuscript. The insightful comments helped us to substantially improve our study.

Reviewer 2 Report

Comments and Suggestions for Authors

Surgery on the lung often leads to worse prognoses because it is believed that local anesthesia suppresses lung function. The authors of this study investigated not only the physical properties of anesthesia but also its biological effects on lung cancer using ropivacaine and lung adenocarcinoma A549 cells. Specifically, they observed changes in the expression pattern of ACE2, a common marker for lung cancer diagnosis, and other related genes. Ultimately, they concluded that ropivacaine administration inhibited A549 cell biology in conjunction with ACE2 upregulation via inhibition of the WNT1 (wingless/Integrated 1) pathway.

#### Review Comments:

The data are well-collected, but the expression of the work needs more sophistication, as follows:

**Major Points:**

1. The introduction on why anesthesia is problematic for lung cancer is insufficient in Lines 34 and 35. Explain what the issues are and why discussions about the effects of anesthesia have arisen.

2. In section 2.1.1, define treatment groups (e.g., C, R1 ~R6, etc.) before explaining the figure.

3. In section 2.1.1, Line 95, if siRNA is used, the knockdown efficiency should be confirmed and calculated first, using RT-PCR and immunoblotting (even if other figures reference this later).

4. In section 2.2.3, if Bland-Altman analyses are applied, the authors should provide a point-to-point discussion about random and systematic errors, rather than just noting that “there was no significant difference in the results between these two methods.”

5. In immunofluorescent analyses, the methodology describing the time schedule is insufficient. Clarify the time course for siRNA, ropivacaine treatment, and measurements. The figure legend indicates ropivacaine treatment for 2 hours, while the M&M section mentions 24 hours. This relates to the longevity of siRNA because ACE2 proteins increase even under RNAi on ACE2. This point should also be addressed in the Discussion section.

6. Reference [25] mentions that ACE2 expression in lung tissue from patients is increased, raising the risk of infectious diseases such as COVID-19. Discuss this perspective in the context of cancer treatment—is it ambivalent or not?

7. Even if the authors acknowledge the limitations of this work, if they wish to generalize the results, at least two cell lines should be tested to support the primary findings. If not, the title should be changed to “Ropivacaine administration suppressed A549 lung adenocarcinoma cell proliferation and migration via ACE2 upregulation and inhibition of the Wnt1 pathway.”

8. In the Discussion section, if current data are used for discussion, the corresponding figure should be noted. For example, “The present study demonstrated that two hours was a sufficient exposure duration for the anti-cancer effects of ropivacaine to be demonstrated (Fig. XX).”

**Minor Points:**

1. References are misaligned. Reference [1] appears in the Copyright section.

2. There are inconsistencies in the spelling of WNT1/wnt1. Generally, "Wnt1" is used when referring to the pathway, and "WNT1" when talking about the gene. If you have a specific distinction in mind, please explain.

3. The English expression needs improvement. For example, “4.2. Anesthetic administration Ropivacaine was obtained from the R group,” may be a miswriting. It is recommended that this manuscript be checked by a native English speaker.

Comments on the Quality of English Language

The English expression needs improvement. For example, “4.2. Anesthetic administration Ropivacaine was obtained from the R group,” may be a miswriting. It is recommended that this manuscript be checked by a native English speaker.

Author Response

(The authors gave the same response as above.)

Round 2

Reviewer 2 Report

Comments and Suggestions for Authors

The authors responded to the reviewers' comments properly. It will be useful knowledge in this field. Lastly, although the MDPI office editor may correct it, please check all gene symbols are in italics.